# Multilevel analysis of unhealthy bodyweight among women in Malawi: Does urbanisation matter?

Rotimi Felix Afolabi[1,2]*, Martin Enock Palamuleni[2]

1 Department of Epidemiology and Medical Statistics, Faculty of Public Health, College of Medicine, University of Ibadan, Ibadan, Nigeria, 2 Population Studies and Demography Programme & Population and Health Research Entity, Faculty of Humanities, North-West University, Mmabatho, South Africa

* rotimifelix@yahoo.com

**Data Availability Statement:** The 2015-16 MDHS dataset used for this analysis is readily available and can be access by interested researchers after seeking permission to use the dataset via the DHS website (https://dhsprogram.com/data).

## Abstract

### Background

Underweight and overweight constitute unhealthy bodyweight and their coexistence is symptomatic of the dual burden of malnutrition (DBM) of high public health concern in many sub-Saharan Africa countries. Little is known about DBM and its correlates in Malawi, a country undergoing urbanisation. The study examined net effects of urban residence on unhealthy weights amidst individual- and community-level factors among women in Malawi.

### Methods

Data on 7231 women aged 15–49 years nested within 850 communities extracted from 2015–16 Malawi Demographic and Health Survey were analysed. Women's weight status measured by body mass index, operationally categorised as underweight, normal and overweight, was the outcome variable while urban-rural residence was the main explanatory variable. Multilevel multinomial logistic regression analysis was employed at 5% significant level; the relative-risk ratio (RR) and its 95% confidence interval (CI) were presented.

### Results

Urban residents had a significantly higher prevalence of overweight than rural (36.4% vs. 17.2%; p< 0.001) but a -non-significant lower prevalence of underweight (6.2% vs. 7.4%; p = 0.423). Having adjusted for both individual- and community-level covariates, compared to rural, living in urban (aRR = 1.25; CI: 1.02–1.53) accounted for about 25% higher risk of being overweight relative to normal weight. Higher education attainment, being married and belonging to Chewa, Lomwe or Mang'anja ethnic group significantly reduced the risk of being underweight but heightened the risk of being overweight. Being older and living in wealthier households respectively accounted for about 3- and 2-times higher likelihood of being overweight, while breastfeeding (aRR = 0.65; CI: 0.55–0.76) was protective against overweight. Living in communities with higher poverty and higher education levels reduced and increased the risk of being overweight, respectively. Evidence of community's variability

**Funding:** The author(s) received no specific funding for this work.

**Competing interests:** The authors have declared that no competing interests exist.

in unhealthy weights was observed in that 11.1% and 3.0% respectively of the variance in the likelihood of being overweight and underweight occurred across communities.

## Conclusions

The study demonstrated association between urban residence and women overweight. Other important associated factors of overweight included breastfeeding, community education- and poverty-level, while education attainment, marital status and ethnicity were associated with the dual unhealthy weight. Thus, both individual- and community-level characteristics are important considerations for policy makers in designing interventions to address DBM in Malawi.

## Introduction

The coexistence of underweight and overweight in the same population indicates a dual burden of malnutrition (DBM) of high public health importance. According to World Health Organisation report, 462 million and 1.9 billion adults respectively were underweight and overweight globally [1]. Worrisome, there has been a continuous rise in overweight coupled with the unabated underweight prevalence in the last few decades [2, 3]. The global prevalence of underweight among adults marginally reduced from 14% in 1975 to 9% in 2016 [4]; nonetheless, as an indicator of undernutrition, underweight remains a critical public health challenge especially in low- and middle-income countries [5]. On the other hand, the prevalence of overweight among adults has increased from 20% in 1975 to 39% in 2016 globally [3, 4].

The dual burden of unhealthy weight has become a common characteristic of many countries, affecting the vulnerable population especially the women and their children [6]. These unhealthy weights among women of childbearing age have been implicated as critical risk factors of morbidity and mortality. For instance, being underweight increases the risk of diseases and adverse maternal and child health conditions; it also increases the risk of dying largely occasioned by external causes [7–10]. Similarly, previous studies have associated spectrum of adverse pregnancy outcomes including maternal and child deaths with overweight or obese [2, 3, 6, 7]. The rising burden of unhealthy bodyweights, which translates to increasing levels of mortality and morbidity, has made DBM a global health priority.

The DBM existence poses a serious developmental threat to many sub-Saharan Africa countries known for extremely high rates of malnutrition [11]. Maternal/child undernutrition and overweight accounted for 826,204 and 266,768 deaths, respectively [12]. Of 821.6 million people undernourished globally, nearly one-third lived in sub-Saharan Africa including Malawi [13]. According to USAID [14], Malawi is a developing country experiencing DBM. The DBM may occur at individual level (co-occurrence of overweight and vitamin/mineral deficient such as anaemia in the same individual), household level (maternal overweight and child underweight co-occurring within the same household) and population level (co-existence of overweight and underweight in the same community or country). Even though a few studies have been conducted on DBM at the individual level [15, 16], household level [17] and combined levels [18], only a single-level analytical cross-sectional study among Dedza district women has occurred at the population level [19] in Malawi. The findings, however, could not be generalised to the Malawian population as it is limited in scope. Even though few nutrition policies and strategies have been initiated and implemented, prevalence of underweight (9% -

1992 to 7% - 2016) declined rather marginally and overweight (10% - 1992 to 21% - 2016) increased persistently over the last two decades in Malawi [20].

Malawi is a country undergoing a nutrition transition; however, nutrition transition theory has linked urbanisation with DBM. Several studies [21–24] have shown that urbanisation is associated with DBM. Many of these studies clearly attest to individual setting or country peculiarities, though with some similarities. The similar pattern is that residing in urban increases the risk of being overweight but lowers the risk of being underweight. Moreover, literature has documented that health outcomes—including unhealthy weights—are often influenced both by individual- and community-level characteristics. For instance, a strong association between urban residence and DBM has been postulated to be explained by individual- and community-level socio-economic characteristics in low- and middle-income countries [25]. Literature is replete on how DBM at population level is influenced by individual- and community-level factors simultaneously [24, 26–30]. However, empirical evidence on the contextual effects on women dual unhealthy weight is yet to be fully documented in Malawi.

In light of this, using large and nationally representative data, it is necessary to further explore the DBM in a low-income setting like Malawi where rapid urban growth poses clear and growing challenges [14, 16, 31]. Information on unhealthy weight and its risk factors could provide evidence-based knowledge that may inform malnutrition prevention efforts by health administrators. This study, therefore, employed multilevel multinomial logistic regression (MMLR) analysis to investigate the effect of urban residence on coexistence of underweight and overweight among women of childbearing age while controlling for other background characteristics. The multilevel method was applied to account for data hierarchical structure and random residual components often associated with a large dataset. This is important for unbiased inferences [32] on risk factors associated with women's unhealthy body weight measured by body mass index (BMI). By and large, the result of this study could ascertain key predictors that can be targeted to prevent overweight and obesity while keeping underweight under control in Malawi and other similar settings.

## Methods

### Study design and sampling procedure

The study extracted data from 2015–16 Malawi Demographic and Health Survey (MDHS). The survey contains information on anthropometric measures that assess nutritional status for all eligible women aged 15–49 years, among others. The survey utilised a two-stage cluster sampling design using the sampling frame produced for the 2008 Malawi Population and Housing Census, as provided by the Malawi National Statistical Office. The sampling frame contains the list of enumeration areas. These are the primary sampling units known as clusters, in which 850 (173—urban; 677—rural) were sampled at the first stage. At the second stage, 63 (30—urban; 33—rural) clusters were selected as the secondary sampling units which amount to 27,516 households sampled. Of these, 26,361 households were interviewed. The detailed sampling design and procedure has been reported in the 2015–16 MDHS report [20].

### Study population and variables

The study data focused on nutritional status of women of reproductive age. As such, they contained variables on women's characteristics as well as detailed information on anthropometric measures on height and weight used to calculate several measures of nutritional status including the BMI. However, women with missing information or 'don't know' records were excluded from the analysis. Also, women who were pregnant at the time of the survey together with those who recently gave birth in the last two months were excluded from the analysis. Fig 1 presents

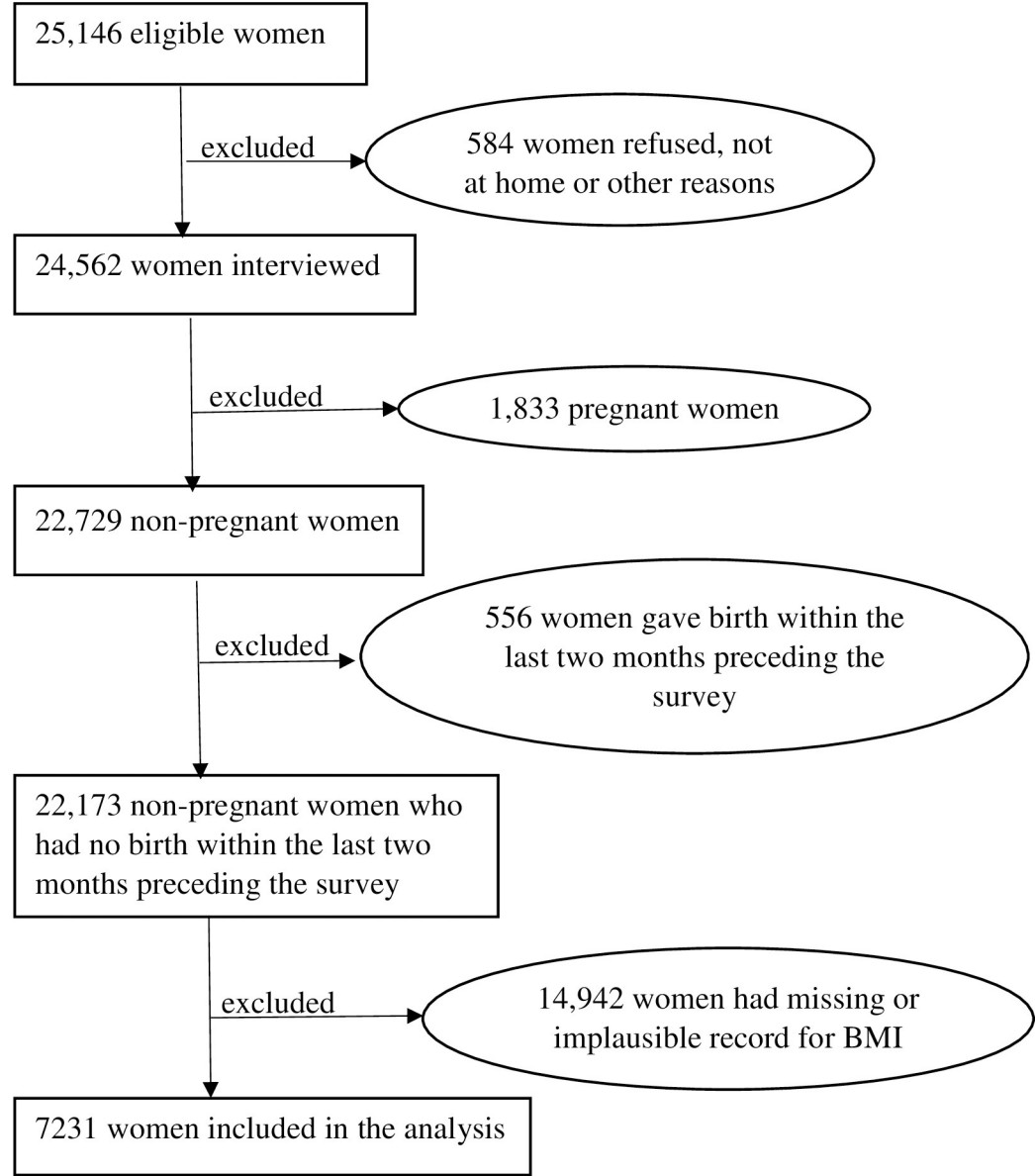

**Fig 1. Flowchart depicting the study population's selection procedure.**

the study population selection details. Data on 7,231 women (level 1) nested within 850 communities (level 2) were weighted and analysed for this study. In the present study, the term 'cluster' is used interchangeably with 'community' to indicate where women live. This describes similarities or clustering within the same geographical living environment due to the idea of sharing a common primary sampling unit.

**Dependent variable.** The outcome variable for this study was women's weight status measured using BMI. The BMI is computed as the ratio of weight in kilograms (kg) to height in meters ($m^2$) and categorised as underweight (BMI<18.5 kg/$m^2$), normal weight (18.5$\leq$BMI$\geq$24.9 kg/$m^2$) and overweight (BMI$\geq$25.0 kg/$m^2$). This, in the course of data analysis, was coded as follows: normal weight = 0, underweight = 1 and overweight = 2 using

normal (healthy) weight as the reference category. Henceforth, overweight/obese shall be referred to as overweight while underweight or overweight as unhealthy weight in this study.

**Independent variables.** The main explanatory variable for this study was urban-rural residence. Other explanatory variables controlled for in this analysis were selected with guidance of empirical literature [33] and categorised into individual-level and community-level factors as appropriate. Current age, household wealth index, the highest educational attainment, employment, marital status, age at first-birth, parity, current breastfeeding, contraceptive use, tobacco use, and media exposure were included to define individual-level factors. Meanwhile, region, ethnicity, poverty level (proportion of women residing in households below the poverty level, 40% of the wealth index in the community), education level (proportion of women who have at least secondary level of education in the community) and media exposure level (proportion of women who have exposure to newspaper/magazine, radio or television in the community) were included to define community-level factors.

The following variables are recoded, for the purpose of analysis, as follows—current age: 1 (15–24years), 2 (25–34years) and 3 (35–49years); age at first-birth: 0 (no birth), 1 (<20 years) and 2 (≥20 years); wealth quintile: 0 (poor), 1 (middle) and 2 (rich); marital status: 0 (never married), 1 (currently married) and 2 (formerly married); parity: 0 (zero child), 1 (1–2 children) and 2 (≥3 children); contraceptive use: 0 (none), 1 (oral/pill) and 2 (others); tobacco use: 0 (not using) and 1 (using); media exposure: 0 (not exposed) and 1 (exposed). Meanwhile, region (northern, central, southern) and ethnicity (Chenwa, Tumbuka, . . ., other) were not recoded. Both poverty level and education level were respectively divided into tertiles (low = 0, medium = 1 and high = 2), while media exposure level was categorised as low = 0 and high = 1.

## Statistical data analysis

The study employed descriptive statistics at the univariate level. The Chi-square test was used to examine association (*t* test, for difference in means where applicable) between urban-rural residence and selected characteristics. Owing to the hierarchical nature of the data and polychotomous nature of the outcome variable, crude and adjusted MMLR models were applied to examine the association between unhealthy weight and the characteristics at bivariate and multivariable levels, respectively. All factors significantly (p<0.05) associated with either underweight or overweight at bivariate level were included in the final model.

**Model description.** The multilevel model has the potential of handling data of hierarchical structure and it is more efficient than traditional regression methods even at a relatively few clustering in groups [32, 34]. Considering individuals in households nested in a community, individual-level (level-1) and community-level (level-2) variables are to predict women's unhealthy weight in this study. The MMLR model is an extension of the binary logistic model expressed as:

$$log_e(\frac{\pi(y_{ij}=k)}{1-\pi(y_{ij}=k)}) = \beta_{00} + \beta_{11}X_{1ij} + \cdots + \beta_{1p}X_{pij} + \beta_{21}Z_{1j} + \cdots + \beta_{2q}Z_{qj} + \omega_{0j} + \theta_{ij} \quad (1)$$

where

$y_{ij}$–stands for each unhealthy weight for i[th] individual in the j[th] community
$\pi(y_{ij}=k)$—stands for the probability of having k[th] unhealthy weight ($k = 1,2$ such that heathy weight is the reference category)
$(\beta_{00}+\omega_{0j})$—called random intercept for the j[th] community

$\beta_{00}$–random intercept for all the communities which represents the log odds of being underweight or overweight relative to normal weight when all the covariate variables in the model are evaluated at zero

$\omega_{0j} \sim N(0, \sigma^2_{\omega_0})$ is the random error term for the j$^{th}$ community across all the communities such that level-2 error variance is $\sigma^2_{\omega_0}$

$\theta_{ij} \sim N(0, \sigma^2_e)$ is the residual effect (variation) such that level-1 error variance is $\sigma^2_e$

$\beta_{1i}$; $i = 1, \cdots, p$ is the regression coefficient corresponding to level-1 covariates ($X_{ij}$); it captures the change in the probability of being underweight or overweight per unit change in the level-1 i$^{th}$ covariate.

$X_{pij}$–represents p$^{th}$ predictor variable for the i$^{th}$ individual in the j$^{th}$ community

$\beta_{2i}$; $i = 1, \cdots, q$ is the regression coefficient corresponding to level-2 covariates ($Z_j$)

$Z_{qj}$—stands for q$^{th}$ predictor variable measured at j$^{th}$ community-level

Eq (1) denotes the relative probability of a woman in the i$^{th}$ household nested in the j$^{th}$ community having k$^{th}$ unhealthy weight. Meanwhile, by converting Eq (1) to the probability of a woman having k$^{th}$ unhealthy weight yields Eq (2)

$$\pi(y_{ij} = k) = \frac{\exp(\beta_{00} + \beta_{11}X_{1ij} + \cdots + \beta_{1p}X_{pij} + \beta_{21}Z_{1j} + \cdots + \beta_{2q}Z_{qj} + \omega_{0j} + \theta_{ij})}{1 + \sum_k(\beta_{00} + \beta_{11}X_{1ij} + \cdots + \beta_{1p}X_{pij} + \beta_{21}Z_{1j} + \cdots + \beta_{2q}Z_{qj} + \omega_{0j} + \theta_{ij})} \quad (2)$$

**Modelling approach.** In all, four models were fitted to investigate the effect of urban residence adjusted for selected individual- and community-level factors on unhealthy weight. The null Model-0 with no covariate was fitted to account for the extent of variance that existed between individual- and community-level effects. Model-1 and Model-2 respectively included individual-level and community-level variables; and Model-3 included all the significant variables in Model-1 and Model-2.

In the present study, fixed effects were presented using relative-risk ratios (RR) with the associated 95% confidence intervals (CI) and/or p-values. Meanwhile, random effects were summarised using intra-cluster correlation coefficient (ICC) and proportional change in variance (PCV). The ICC is expressed as

$$ICC = \frac{\sigma^2_{\omega_m}}{\sigma^2_{\omega_m} + 3.29}; \quad m = 0, 1, 2, 3 \text{ models} \quad (3)$$

Eq (3), usually expressed in percentage, explains the magnitude of exposure by which women in the same community are exposed to the same characteristics associated with unhealthy weight. Simply put, it presented a similarity measure (variation) of relative risks of being underweight or overweight in the same community. An ICC of at least 2% suggests an important cluster level effect that requires a multilevel analysis [35]. The PCV indicates the magnitude to which the addition of predictors to the null model would better explain the risk of being underweight or overweight relative to the normal weight. This is expressed as

$$PCV = \frac{(\sigma^2_{\omega_0} - \sigma^2_{\omega_m})}{\sigma^2_{\omega_0}}; \quad m = 1, 2, 3 \text{ models} \quad (4)$$

Lastly, model fit was investigated using Akaike information criterion (AIC) which suggests the smaller the value the better the model fits. All analyses were conducted at 5% level of significance using Stata MP version 14.0.

### Ethical approval

This study was premised on the analysis of secondary analysis of the Demographic and Health Surveys data. The National Health Sciences Research Committee Malawi and the ICF Institutional Review Board reviewed and approved the survey protocol. At the time of the survey, all participants were made to sign written agreement form prior to the interview while all data collection and measurement activities were conducted in strict confidence. Kindly refer to the 2015–16 MDHS report [20] for the details of the ethical approval. Asides, the authors obtained permission from the data owners to use the dataset for the present analysis.

## Results

### Women's characteristics and its association with urban residence

Most women were aged 15–24 years (41.3%), currently married (63.5%), and lived in a wealthier household (43.3%) or Southern region (46.2%). Nearly three-quarters of the women were currently not breastfeeding and had no or primary level of education. Most women belonged to communities with low education (36.4%), medium poverty (37.7%) or high level of media exposure (67.3%). All variables considered, but region of residence, have significant relationship with urban-rural residence (p<0.05). Of note, 74.1% of women who attained higher-level of education lived in urban; 59.8% and 64.7% of communities respectively with high education and with low poverty level were urban residents (Table 1).

### Patterns of unhealthy weight status by urban-rural residence

Fig 2 depicts the percentage distribution of the women according to their unhealthy bodyweight status by urban-rural residence. The finding noted a coexistence of underweight (7.2%) and overweight (20.7%) among Malawian women. Higher prevalence of overweight (36.2%) but marginally lower underweight (6.2%) was observed among urban residents respectively compared with overweight (17.2%) and underweight (7.4%) in rural. Although not presented, the association between urban and rural underweight ($\chi$ = 0.6433; p = 0.423) was statistically non-significant while that of overweight ($\chi$ = 29.4845; p<0.001) was significant.

### Multilevel modelling of unhealthy bodyweight

**Crude model.** The association of each background characteristic with unhealthy bodyweights, without taking into consideration the effects of other variables, is presented in Table 1. Among all the factors considered, neither tobacco use nor media exposure (both at individual- and community-level) was significantly associated with the likelihood of having unhealthy weight. Other characteristics considered were significantly related to being overweight (p<0.05) and being underweight (p<0.05), except for household wealth (both at individual- and community-level). For instance, urban women were about 14% times less likely to be underweight (RR: 0.86; CI: 0.66–1.10)—though statistically non-significant, and 171% times more likely to be overweight (RR: 2.71; CI: 2.33–3.15) over healthy weight compared to their rural counterparts. The tendency of being underweight was significantly lower among women aged 25–34 years (RR: 0.70; CI: 0.56–0.88), who had attained higher education or lived in communities with a high proportion of education; the risk of being overweight was significantly higher among women aged 25–34 years (RR: 3.14; CI: 2.68–3.67), who had attained higher education or lived in communities with a high proportion of education. Notably, the likelihood of being underweight (RR: 0.66; CI: 0.53–0.83) or overweight (RR: 0.58; CI: 0.50–0.67) was significantly lower among women who were currently breastfeeding (Table 1).

**Table 1. Distribution of participants by urban residence and unhealthy bodyweight association with individual- and community-level characteristics.**

| Characteristics | Total | Urban | Underweight | Overweight |
|---|---|---|---|---|
| | n(%) | % | RR (CI) | RR (CI) |
| *Individual-level* | | | | |
| **Current age** | | <0.001*** | | |
| 15–24 (R) | 2964(41.0) | 19.3 | 1 | 1 |
| 25–34 | 2258(31.2) | 21.3 | 0.70(0.56,0.88)** | 3.14(2.68,3.67)*** |
| 35–49 | 2009(27.8) | 14.2 | 0.81(0.65,1.02) | 3.85(3.28,4.52)*** |
| mean±sd | 28.4±9.4 | 27.5±8.9* | 25.9±9.4 | 32.0±8.5 |
| **Wealth** | | <0.001*** | | |
| Poor (R) | 2526(34.9) | 1.9 | 1 | 1 |
| Middle | 1329(18.4) | 4.8 | 0.98(0.76,1.27) | 1.56(1.29,1.89)*** |
| Rich | 3376(46.7) | 39.1 | 0.93(0.76,1.15) | 3.24(2.79,3.76)*** |
| **Education** | | <0.001*** | | |
| No education (R) | 853(11.8) | 4.6 | 1 | 1 |
| Primary | 4315(59.7) | 10.8 | 1.02(0.76,1.36) | 0.79(0.65,0.95)* |
| Secondary | 1862(25.8) | 37.9 | 0.93(0.67,1.29) | 1.22(0.99,1.51) |
| Higher | 201(2.8) | 74.1 | 0.19(0.05,0.78)* | 2.26(1.58,3.23)*** |
| **Employment** | | <0.001*** | | |
| Not working (R) | 2727(37.7) | 23.2 | 1 | 1 |
| Working | 4504(62.3) | 15.8 | 0.75(0.62,0.91)** | 1.38(1.21,1.57)*** |
| **Marital status** | | <0.036* | | |
| Never married (R) | 1662(23.0) | 26.9 | 1 | 1 |
| Currently married | 4554(63.0) | 16.2 | 0.44(0.35,0.54)*** | 2.72(2.28,3.25)*** |
| Formerly married | 1015(14.0) | 15.6 | 0.59(0.44,0.79)*** | 2.43(1.94,3.04)*** |
| **Age at first birth** | | <0.001*** | | |
| No birth (R) | 1607(22.2) | 25.0 | 1 | 1 |
| <20 | 3886(53.7) | 14.2 | 0.47(0.38,0.58)*** | 2.33(1.95,2.78)*** |
| ≥20 | 1738(24.0) | 22.2 | 0.58(0.45,0.75)*** | 2.68(2.21,3.26)*** |
| **Parity** | | <0.001*** | | |
| 0 (R) | 1607(22.2) | 25.0 | 1 | 1 |
| 1–2 | 2136(29.5) | 23.3 | 0.48(0.37,0.61)*** | 1.88(1.55,2.28)*** |
| ≥3 | 3488(48.2) | 12.6 | 0.52(0.42,0.65)*** | 2.90(2.42,3.48)*** |
| **Breastfeeding** | | <0.001*** | | |
| No (R) | 5347(73.9) | 20.4 | 1 | 1 |
| Yes | 1884(26.1) | 13.2 | 0.66(0.53,0.83)*** | 0.58(0.50,0.67)*** |
| **Contraceptive use** | | 0.032* | | |
| Not using (R) | 3587(49.6) | 19.5 | 1 | 1 |
| Oral (pill) | 154(2.1) | 25.7 | 0.24(0.07,0.76)* | 2.00(1.37,2.91)*** |
| Other methods | 3490(48.3) | 17.2 | 0.61(0.50,0.74)*** | 1.43(1.26,1.61)*** |
| **Tobacco use** | | <0.001*** | | |
| Not using (R) | 7178(99.3) | 18.6 | 1 | 1 |
| Using | 53(0.7) | 3.9 | 1.41(0.55,3.63) | 0.95(0.46,1.94) |
| **Media exposure** | | 0.004** | | |
| No exposure (R) | 348(4.8) | 11.6 | 1 | 1 |
| Has exposure | 6883(95.2) | 18.9 | 1.05(0.68,1.62) | 1.08(0.81,1.44) |
| *Community-level* | | | | |
| **Residence** | | | | |
| Rural (R) | 5636(77.9) | | 1 | 1 |

(*Continued*)

**Table 1.** (Continued)

| Characteristics | Total | Urban | Underweight | Overweight |
|---|---|---|---|---|
| | n(%) | % | RR (CI) | RR (CI) |
| Urban | 1595(22.1) | | 0.86(0.66,1.10) | 2.71(2.33,3.15)*** |
| **Ethnicity** | | <0.001*** | | |
| Chewa | 2176(30.1) | 12.2 | 2.72(1.09,6.76)* | 0.49(0.35,0.70)*** |
| Tumbuka | 748(10.3) | 24.1 | 2.26(0.87,5.85) | 0.72(0.50,1.04) |
| Lomwe | 1352(18.7) | 19.9 | 3.27(1.30,8.19)* | 0.52(0.36,0.75)*** |
| Tonga | 268(3.7) | 21.2 | 3.05(1.11,8.42)* | 0.58(0.36,0.93)* |
| Yao | 835(11.5) | 18.7 | 2.85(1.12,7.28)* | 0.55(0.37,0.80)** |
| Sena | 318(4.4) | 16.3 | 2.36(0.86,6.46) | 0.44(0.28,0.70)*** |
| Nkhonde | 94(1.3) | 38.0 | 2.74(0.80,9.42) | 0.88(0.48,1.59) |
| Ngoni | 905(12.5) | 26.2 | 3.10(1.22,7.90)* | 0.75(0.52,1.09) |
| Mang'anja | 166(2.3) | 29.8 | 3.83(1.35,10.85)* | 0.48(0.28,0.83)** |
| Nyanga | 148(2.0) | 17.4 | 2.16(0.66,7.07) | 0.83(0.50,1.38) |
| Other (R) | 221(3.1) | 20.1 | 1 | 1 |
| **Region** | | 0.482 | | |
| Northern (R) | 1386(19.2) | 19.2 | 1 | 1 |
| Central | 2475(34.2) | 19.2 | 1.10(0.82,1.47) | 0.67(0.55,0.82)* |
| Southern | 3370(46.6) | 17.7 | 1.35(1.03,1.77)* | 0.65(0.53,0.79)* |
| **Poverty level** | | <0.001*** | | |
| Low (R) | 2474(34.2) | 64.7 | 1 | 1 |
| Medium | 2534(35.0) | 1.6 | 1.13(0.89,1.44) | 0.40(0.35,0.47)*** |
| High | 2223(30.7) | 0.1 | 1.12(0.88,1.43) | 0.30(0.26,0.36)*** |
| **Education level** | | <0.001*** | | |
| Low (R) | 2379(32.9) | 0.2 | 1 | 1 |
| Medium | 2412(33.4) | 4.3 | 0.87(0.70,1.08) | 1.50(1.26,1.78)*** |
| High | 2440(33.7) | 59.8 | 0.77(0.61,0.98)* | 3.31(2.81,3.90)*** |
| **Media exposure level** | | 0.006** | | |
| Low (R) | 2465(34.1) | 11.9 | 1 | 1 |
| High | 4766(65.9) | 21.7 | 1.09(0.89,1.33) | 1.15(0.98,1.35) |

* p<0.05

** p<0.01

*** p<0.001; sd–standard deviation; RR(CI)–crude relative-risk ratio (95% confidence interval); R–reference category

**Adjusted models' fixed effects.** Table 2 presents the results of the four models which include individual- and community-level fixed effects along with random effects. Considering the favoured (final) model, only the highest educational attainment, marital status and ethnicity remained significantly associated with the relative risk of being underweight over healthy weight (panel 1; Model-3). The risk of being underweight decreased with increasing highest educational attainment such that the risk of underweight was 86% lower among women who had higher education (aRR: 0.14; CI: 0.03–0.59) than no formal education. Similarly, ever married women (currently—aRR: 0.37; CI: 0.27–0.50; formerly—aRR: 0.46; CI: 0.32–0.66) had lesser risks of being underweight relative to healthy weight. However, Chewa (aRR: 2.53; CI: 1.00–6.37), Lomwe (aRR: 3.01; CI: 1.19–7.60), Yao (aRR: 2.68; CI: 1.04–6.89), Ngoni (aRR: 3.06; CI: 1.19–7.86) and Mang'anja (aRR: 3.56; CI: 1.24–10.19) women were more likely to be underweight.

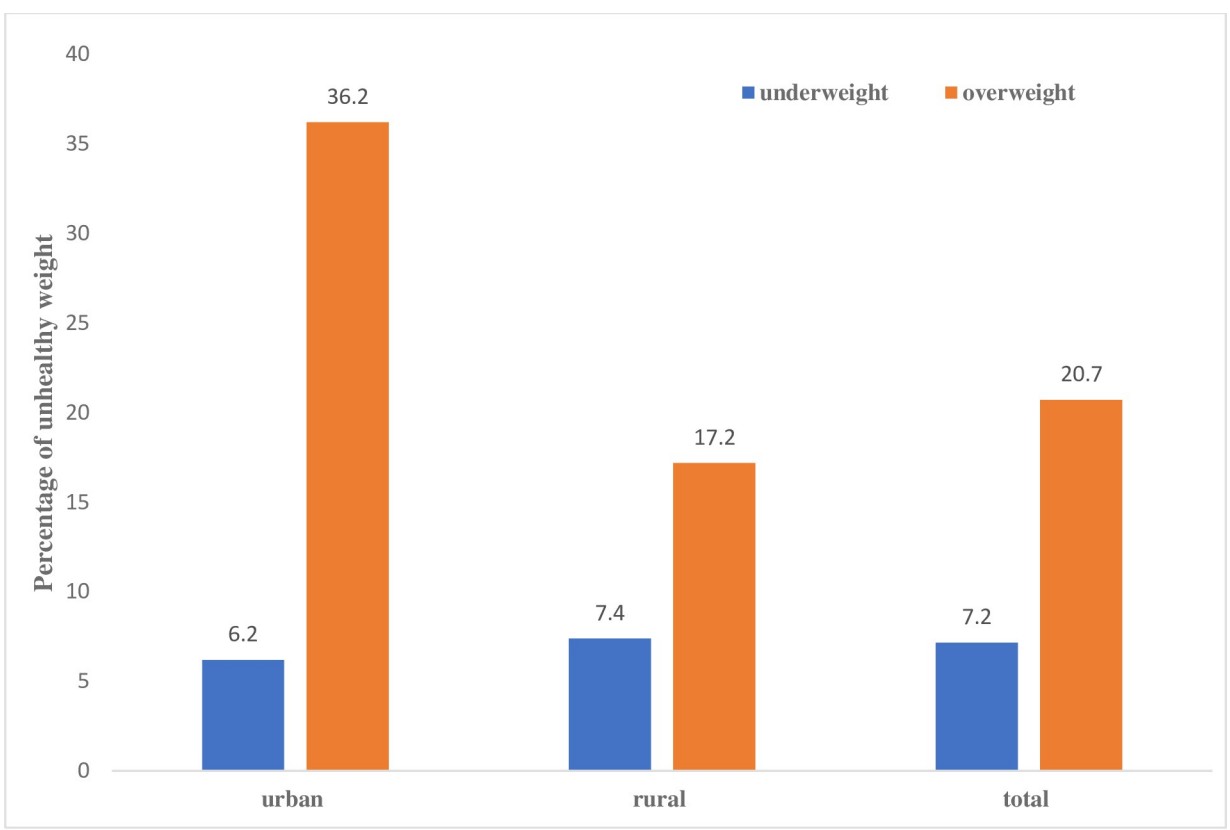

**Fig 2. Percentage distribution of unhealthy weight by urban-rural residence.**

Women's age, wealth index, education attainment, marital status, currently breastfeeding, ethnicity, and community poverty- or educational-level were significantly associated with risks of being overweight (panel 2; Model-3). Compared to younger women, women aged 25–34 (aRR = 2.39; CI: 1.98–2.87) and 35–49 (aRR = 2.82; CI: 2.30–3.45) years respectively had about 2- and 3-times higher risks of being overweight relative to healthy weight. While women who were currently breastfeeding (aRR = 0.65; CI: 0.55–0.76) had lesser risks, those who were currently married (aRR = 1.90; 1.52–2.38) had greater risks of being overweight. Women who had attained higher education (aRR = 1.49; 1.02–2.18) and resided in communities with high education level (aRR = 1.68; CI: 1.31–2.16) respectively had about 50% and 70% higher risks of being overweight. While women who resided in rich households (aRR = 2.05; CI: 1.71–2.46) were more likely to be overweight relative to healthy weight, those who lived in communities of households with high poverty level (aRR = 0.65; CI: 0.55–0.76) and belonged to Chewa, Tumbuka, Lomwe, Tonga, Sena or Mang'anja ethnicity were protective against the risk of being overweight (panel 2; Model-3).

**Adjusted model's random effects.** In Table 2, Model-0 showed a statistically significant joint variance of underweight and overweight ($\sigma_{w12}$ = −0.119; $p$ = 0.038) across the communities (not presented). Although variation in underweight between communities was non-significant in all models, variation in the risk of being overweight was statistically significant even after adjusting for individual-level covariates (p<0.05). As indicated in Model-0, ICC values of 11.1% and 3.0% respectively indicate the variations in overweight and underweight explained by the differences between communities; however, both values reduced to ICC = 2.3% in the final model. All ICC ≥ 2% confirm the adequacy of the multilevel method. Although

**Table 2. Effects of individual- and community characteristics on women's unhealthy bodyweight in Malawi.**

| Characteristics | Underweight Model 1 | | | | Overweight | | | |
|---|---|---|---|---|---|---|---|---|
| | Model-0 | Model-1 | Model-2 | Model-3 | Model-0 | Model-1 | Model-2 | Model-3 |
| | | aRR (CI) | aRR (CI) | aRR (CI) | | aRR (CI) | aRR (CI) | aRR (CI) |
| *Fixed effects* | | | | | | | | |
| *Individual-level* | | | | | | | | |
| **Age** | | | | | | | | |
| 15–24 (R) | | 1 | | 1 | | 1 | | 1 |
| 25–34 | | 1.15(0.81,1.65) | | 1.18(0.89,1.58) | | 2.57(2.07,3.19)*** | | 2.39(1.98,2.87)*** |
| 35–49 | | 1.19(0.77,1.82) | | 1.28(0.94,1.76) | | 3.12(2.41,4.03)*** | | 2.82(2.30,3.45)*** |
| **Wealth** | | | | | | | | |
| Poor (R) | | 1 | | 1 | | 1 | | 1 |
| Middle | | 0.96(0.75,1.25) | | 0.94(0.73,1.23) | | 1.45(1.19,1.77)*** | | 1.37(1.12,1.68)** |
| Rich | | 0.89(0.70,1.12) | | 0.89(0.69,1.15) | | 2.80(2.37,3.30)*** | | 2.05(1.71,2.46)*** |
| **Education** | | | | | | | | |
| No education (R) | | 1 | | 1 | | 1 | | 1 |
| Primary | | 0.93(0.68,1.26) | | 0.92(0.68,1.26) | | 1.01(0.82,1.23) | | 0.91(0.74,1.12) |
| Secondary | | 0.68(0.47,1.00)* | | 0.74(0.50,1.09) | | 1.53(1.20,1.94)*** | | 1.13(0.89,1.44) |
| Higher | | 0.13(0.03,0.54)** | | 0.14(0.03,0.59)** | | 2.27(1.54,3.33)*** | | 1.49(1.02,2.18)* |
| **Employment** | | | | | | | | |
| Not working (R) | | 1 | | | | 1 | | |
| Working | | 0.94(0.76,1.15) | | | | 0.92(0.80,1.06) | | |
| **Marital status** | | | | | | | | |
| Never married (R) | | 1 | | 1 | | 1 | | 1 |
| Currently married | | 0.43(0.28,0.64)*** | | 0.37(0.27,0.50)*** | | 1.73(1.27,2.36)*** | | 1.90(1.52,2.38)*** |
| Formerly married | | 0.51(0.32,0.82)*** | | 0.46(0.32,0.66)*** | | 1.53(1.08,2.15)* | | 1.56(1.19,2.04)** |
| **Age at first birth** | | | | | | | | |
| No birth (R) | | 1 | | | | 1 | | |
| <20 | | 0.92(0.54,1.56) | | | | 1.03(0.72,1.47) | | |
| ≥20 | | 1.21(0.67,2.19) | | | | 0.87(0.59,1.28) | | |
| **Parity** | | | | | | | | |
| 0 (R) | | 1 | | | | 1 | | |
| 1–2 | | 0.97(0.70,1.35) | | | | 1.11(0.92,1.33) | | |
| ≥3 | | na | | | | na | | |
| **Breastfeeding** | | | | | | | | |
| No (R) | | 1 | | 1 | | 1 | | 1 |
| Yes | | 0.89(0.68,1.15) | | 0.89(0.69,1.14) | | 0.64(0.54,0.75)*** | | 0.65(0.55,0.76)*** |
| **Contraceptive** | | | | | | | | |
| Not using (R) | | 1 | | | | 1 | | |
| Oral (pill) | | 0.35(0.11,1.13) | | | | 1.34(0.89,1.95) | | |
| Other methods | | 0.86(0.68,1.09) | | | | 1.09(0.95,1.26) | | |
| *Community-level* | | | | | | | | |
| **Residence** | | | | | | | | |
| Rural (R) | | | 1 | 1 | | | 1 | 1 |
| Urban | | | 0.91(0.64,1.30) | 0.92(0.64,1.32) | | | 1.31(1.08,1.60)** | 1.25(1.02,1.53)* |
| **Ethnicity** | | | | | | | | |
| Chewa | | | 3.02(1.14,8.00)* | 2.53(1.00,6.37)* | | | 0.64(0.44,0.92)* | 0.69(0.49,0.97)* |
| Tumbuka | | | 2.41(0.93,6.26) | 2.28(0.87,5.94) | | | 0.68(0.48,0.96)* | 0.68(0.47,0.98)* |
| Lomwe | | | 3.06(1.14,8.21)* | 3.01(1.19,7.60)* | | | 0.64(0.44,0.94)* | 0.65(0.46,0.92)* |

*(Continued)*

**Table 2.** (Continued)

| Characteristics | Model-0 | Model-1 aRR (CI) | Model-2 aRR (CI) | Model-3 aRR (CI) | Model-0 | Model-1 aRR (CI) | Model-2 aRR (CI) | Model-3 aRR (CI) |
|---|---|---|---|---|---|---|---|---|
| | | | **Underweight Model 1** | | | | **Overweight** | |
| Tonga | | | 3.08(1.12,8.49)* | 2.72(0.98,7.54) | | | 0.58(0.38,0.88)* | 0.61(0.39,0.96)* |
| Yao | | | 2.78(1.02,7.55)* | 2.68(1.04,6.89)* | | | 0.69(0.47,1.03) | 0.74(0.51,1.07) |
| Sena | | | 2.19(0.75,6.41) | 2.14(0.77,5.93) | | | 0.61(0.38,0.98)* | 0.60(0.38,0.94)* |
| Nkhonde | | | 2.85(0.83,9.81) | 2.51(0.72,8.72) | | | 0.73(0.42,1.28) | 0.71(0.40,1.27) |
| Ngoni | | | 3.29(1.22,8.85)* | 3.06(1.19,7.86)* | | | 0.89(0.61,1.30) | 0.92(0.64,1.32) |
| Mang'anja | | | 3.53(1.17,10.67)* | 3.56(1.24,10.19)* | | | 0.59(0.34,1.02) | 0.51(0.30,0.88)* |
| Nyanga | | | 2.10(0.63,6.96) | 1.87(0.57,6.16) | | | 0.85(0.52,1.38) | 0.85(0.52,1.41) |
| Other (R) | | | 1 | 1 | | | 1 | 1 |
| **Region** | | | | | | | | |
| Northern (R) | | | 1 | | | | 1 | |
| Central | | | 0.90(0.57,1.42) | | | | 1.00(0.78,1.28) | |
| Southern | | | 1.10(0.70,1.75) | | | | 0.96(0.74,1.23) | |
| **Poverty level** | | | | | | | | |
| Low (R) | | | 1 | 1 | | | 1 | 1 |
| Medium | | | 0.87(0.62,1.23) | 0.84(0.59,1.21) | | | 0.61(0.49,0.75)*** | 0.77(0.62,0.96)* |
| High | | | 0.80(0.54,1.18) | 0.77(0.51,1.16) | | | 0.54(0.42,0.70)*** | 0.77(0.59,1.01) |
| **Education level** | | | | | | | | |
| Low (R) | | | 1 | 1 | | | 1 | 1 |
| Medium | | | 0.85(0.67,1.08) | 0.85(0.67,1.08) | | | 1.28(1.07,1.52)** | 1.22(1.01,1.47)* |
| High | | | 0.71(0.50,1.02) | 0.70(0.48,1.02) | | | 1.67(1.32,2.11)*** | 1.68(1.31,2.16)*** |
| *Random effect* | | | | | | | | |
| Community residual | 0.1033 | 0.0935 | 0.0738 | 0.0773 | 0.4108*** | 0.1501** | 0.0673 | 0.0766 |
| ICC % | 3.0 | 2.8 | 2.2 | 2.3 | 11.1 | 4.4 | 2.0 | 2.3 |
| PCV % | R | 9.5 | 28.6 | 25.2 | R | 63.5 | 83.6 | 81.4 |
| Fit indices | | Model-0 | | Model-1 | | Model-2 | | Model-3 |
| -2LL | | 10886.0 | | 10075.0 | | 10571.0 | | 9957.0 |
| AIC | | 10896.0 | | 10149.0 | | 10649.0 | | **10067.0** |

\* p<0.05

\*\* p<0.01

\*\*\* p<0.001; aRR(CI)–adjusted relative-risk ratio (95% confidence interval); R–reference category; na—omitted due to collinearity with age at first-birth; LL—log-likelihood

underweight and overweight PCV values were slightly higher in Model-2 than in Model-3, Model-3 having the least AIC = 10,067.0 was a better model which demonstrates that inclusion of individual- and community-level characteristics improved the capability of our model in accounting for the variability in unhealthy bodyweight across communities. Relative to Model-0, 25.2% and 81.4% respectively of the variations in underweight and overweight across all the communities were explained by individual- and community-level characteristics included in the final model (Table 2).

## Discussion

The present study investigated the net effects of urbanisation on unhealthy bodyweight measured by BMI among women in Malawi while controlling for individual- and community-

level characteristics. We found that urban residence considerably impacted on unhealthy bodyweight, especially overweight even after accounting for the effects of individual- and community-level factors. To the best of our knowledge, this study appears to have been the first multilevel examination of factors associated with DBM to define population level among Malawian women using most recent nationally representative data.

In this study, the highest educational attainment, marital status and ethnicity were covariates that concomitantly and significantly influenced underweight and overweight. Other significant correlates including current age, wealth status, currently breastfeeding, and community-level education or poverty rate were identified to be associated with only the risk of being overweight. The variation in the risk of unhealthy weights between communities found in this study aligns with prior studies [27, 28], though the variation was non-significant in underweight risk. This suggests that accounting for the contextual effect in explaining the risk of overweight is more crucial compared to underweight in Malawi.

Consistent with previous studies [22, 29, 36], the findings showed that living in urban significantly increases the risk of being overweight but not underweight. Other related studies [24, 30, 37, 38] have also corroborated the findings of this study that the risk of being overweight is higher among urban residents compared to rural. Perhaps the likely reason could be that most urban residents are less involved in physical activities as documented by other studies [18, 39] in Malawi. It is therefore not unlikely for urban women to gain much weight than their counterparts in rural. Nonetheless, the magnitude of the risk of urban residents being overweight substantially decreased after controlling for individual-level and other community-level effects. This indicates that much of the urban-rural variation in the risk of being overweight is largely driven not only by the individual-level but also by other community-level effects. This aligns with a similar study conducted among low- and middle-income countries [25]. Contrary to our result, some studies [36, 40] conducted among sub-Saharan African and South Asian countries have observed that urban residence was neither associated with underweight nor overweight. The reason for this contrasting result could partially be attributed to geographical and racial differences.

In addition, ever married women were at-risk of being overweight, while those unmarried were at-risk of being underweight in this study. This finding aligns with previous studies [37, 41–43]. This is expected as ever married women, compared to unmarried, are more exposed to pregnancy and childbirth that often lead to hastened hormonal or physiological body-weight changes that may lead to overweight [26, 44]. Besides, literature has linked the rise in overweight to cultural or social perception about ideal body-weight in many sub-Saharan African countries [23]. Women body-weight management is therefore crucial to averting avoidable malnutritional challenges [7, 10, 45]. Literature [46] has also associated unhealthy weight with ethnicity differences as observed in this study. This suggests ethnicity differentials in underweight and overweight may not be limited to the characteristics considered in this study, and may be associated with others like genetic or biological characteristics [43, 46].

Education attainment is a key driving force of malnutrition, mostly among sub-Saharan Africa women [47]. Higher education fosters the risk of being overweight but mitigates the likelihood of being underweight as revealed in this study. The finding aligns with previous studies [21, 26, 29, 41] that suggested that higher education empowers women socially and economically. This apparently presents women with opportunities of professional jobs characterised by sedentariness, a circumstance that likely prompts them to cease breastfeeding early or consume excessive poor calory-rich foods [39, 46, 48, 49]. Even though only 3% of the women had attained higher education, about three-quarters of them were urban residents as observed in this study. Higher education is therefore a feature of urban residence, a potent

catalyst for overweight but protection against underweight as further corroborated by other studies [25, 37, 42].

Generally, household wealth has an interwoven link with other predictors of unhealthy weight. For instance, wealth status usually predicts educational attainment [47]. It may also impact on employment status, contraceptive use or parity [44], though these are non-significant risk factors of unhealthy weight after accounting for other variables' effects in this study. The higher educational level attained may boost employment opportunity and lead to rich wealth status, and consequently influence unhealthy weight. The findings of the higher the household wealth the more the risk of being overweight in this study is consistent with previous studies [19, 33, 38, 43]. Besides, living in a community with a low proportion of poverty or high proportion of education rate increased the risk of being overweight in this study; this aligns with a prior study [25]. Intervention strategies therefore should be focused on education and empowerment as a means of reducing DBM, especially in an economically deprived setting where higher education both at individual- and community-levels remains low as observed in this study.

Effective policy strategies to curb unhealthy weight as one advances in age cannot be over-emphasised. In this study, current age is another significant predictor of overweight. This is consistent with previous literature [23, 36, 38, 50] that claimed older women are more prone to be overweight. This could partly be explained by increasing pregnancy or procreation of children, often accompanied by physiological body changes linked to increasing age [26, 48], especially among ever married women who constitute more than three-quarters of the women population in this study. Other possible explanation for the positive impact of increasing age on unhealthy bodyweight could be due to lower likelihood to breastfeed among older women as corroborated earlier [48, 51]. In this study, breastfeeding is protective against overweight which aligns with prior finding [52]. Coupled with a report of a recent decline in breastfeeding in Malawi [14], promotion of breastfeeding in women nutrition, campaign and intervention should therefore be strengthened.

Of note, this result suggests a potential coexistence of overweight and underweight (though statistically non-significant) among older women, residents of communities with low proportion of poverty, and those who were not breastfeeding after controlling for other confounders. In anticipation of higher risk of unhealthy weight especially overweight irrespective of the residence location as urbanisation increases [38, 53], urgent attention on these vulnerable population settings is therefore required to curtail the negative impacts on women's development in Malawi.

## Limitations

The present study, however, acknowledges some limitations. One, the study design is cross-sectional which limits the potential of making causal inferences. Two, women self-reported data without any means of verification in this study may be a potential avenue of recall bias. Asides being a secondary data analysis, the study precludes the inclusion of some other variables like dietary intake, body fat, physical activity (such variables were not available for the present 2015–16 MDHS dataset). Notwithstanding, the strength of the study lies in the usage of large and nationally representative data that allows generalization of the study findings to the entire women of childbearing age in Malawi. In addition, the strength of the work includes the application of multilevel analysis that accounts for the complex hierarchical survey design.

## Conclusions

The present study demonstrated the association between urban residence and women over-weight but not underweight, suggesting that urbanisation negatively impacts overweight in Malawi. Other important drivers of overweight included breastfeeding, community education- and poverty-level. Meanwhile, factors such as higher education attainment, being married and belonging to Chewa, Lomwe or Mang'anja ethnic group were concomitantly associated with reduced risk of underweight and increased risk of overweight. The study revealed that the influence of the community on unhealthy bodyweight, most importantly overweight, has the capability to vary according to different women's socio-economic and cultural backgrounds. Evidence of potential risks of DBM among older women, those who are not breastfeeding or living in a community with low proportion of poverty was observed. Thus, both individual- and community-level characteristics are important considerations for policy makers in designing interventions aimed at addressing unhealthy bodyweight and subsequently DBM in Malawi. Such policies and interventions should be targeted simultaneously at-risk individuals and at-risk communities to improving healthy bodyweight.

## Acknowledgments

The authors appreciate The DHS program, ICF International, USA for providing us with access to use the data.

## Author Contributions

**Conceptualization:** Rotimi Felix Afolabi, Martin Enock Palamuleni.

**Data curation:** Rotimi Felix Afolabi.

**Formal analysis:** Rotimi Felix Afolabi.

**Methodology:** Rotimi Felix Afolabi.

**Resources:** Rotimi Felix Afolabi, Martin Enock Palamuleni.

**Software:** Rotimi Felix Afolabi.

**Supervision:** Martin Enock Palamuleni.

**Validation:** Rotimi Felix Afolabi.

**Visualization:** Rotimi Felix Afolabi, Martin Enock Palamuleni.

**Writing – original draft:** Rotimi Felix Afolabi.

**Writing – review & editing:** Rotimi Felix Afolabi, Martin Enock Palamuleni.

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
