## [Decision Letter · Decision Letter 0]

3 Feb 2021

PONE-D-20-37443

Multilevel analysis of unhealthy bodyweight among women in Malawi: does urbanisation matter?

PLOS ONE

Dear Dr. Afolabi,

Thank you for submitting your manuscript to PLOS ONE. After careful consideration, we feel that it has merit but does not fully meet PLOS ONE’s publication criteria as it currently stands. Therefore, we invite you to submit a revised version of the manuscript that addresses the points raised during the review process.

We look forward to receiving your revised manuscript.

Kind regards,

Kannan Navaneetham, PhD

Academic Editor

PLOS ONE

Journal Requirements:

Reviewers' comments:

Reviewer's Responses to Questions

**Comments to the Author**

1. Is the manuscript technically sound, and do the data support the conclusions?

Reviewer #1: Yes

Reviewer #2: Yes

2. Has the statistical analysis been performed appropriately and rigorously? 

Reviewer #1: Yes

Reviewer #2: Yes

3. Have the authors made all data underlying the findings in their manuscript fully available?

Reviewer #1: No

Reviewer #2: Yes

4. Is the manuscript presented in an intelligible fashion and written in standard English?

Reviewer #1: Yes

Reviewer #2: Yes

5. Review Comments to the Author

Reviewer #1: The authors have used multi-level multinomial logistic regression analysis to assess the net effects of urban residence on the double burden of malnutrition amidst individual and community level factors among women in Malawi.

Abstract

The authors indicate unhealthy weight, underweight and overweight as symptomatic of the dual burden of DBM. Stating, unhealthy weight, then underweight and overweight at the same time would imply that overweight and underweight are different from unhealthy weight whereas they are both unhealthy weight. I would suggest correction of this sentence. The sentence does not read well.

Under the results in the abstract and the entire paper you use the phrase, 'INSIGNIFICANT' i am not sure if this is the right word particularly when referring to a statistical outcome. I am wondering if 'non-significant' will make the interpretation sound better. Explore which is the best word to use and correct accordingly. You also indicate that ethnicity was a significant risk factor of DBM, can you specify which ethnic group and perhaps put the aRR and CIs. This will help the reader to make a follow-up in the paper, why this particular ethnic group (s) is peculiar. Line 40 please replace the word likely with the word likelihood and insert while before breast feeding.

Background

This section is well written and provides the background, country context and why it is necessary to study net effects of urban residence on DBM in Malawi.

Methods section

This section is also explicitly written. I am not sure about the link between the explanation of the study, objective and the outcome variable. In the title you are talking about unhealthy body weight, in the objective DBM while the outcome variable is BMI. The link between this three is necessary. Although all these concepts are related they do not mean the same thing and there is need for authors to carefully provide consistency in what the study really seeks to address. I don't think expressing the MMLR model in formula is necessary for the reader.I would rather have it as an attachment separate from the main manuscript and describe briefly how models were constructed at the time of data analysis.

Results

The results section is well written/ The conventional way to indicate statistical significance at 95% CI would be *** instead of *. Under the Adjusted model's random effects-line 257, you used abbreviations such as ICC, PCV and AIC values without first explaining their meaning.

Discussion

this section is also generally well written. However, line 316-318 is not necessary because you have earlier stated the objected of the study. In line 342 you indicate that ever married women were at risk of being overweight while un-married were at risk of underweight.In justifying why married women are more overweight you posit that it is because married women are exposed to pregnancy and child birth which lead to hormonal and physiological body-weight changes.One interesting question would be that, are un-married women not exposed to these body changes, or it is that there are low levels of pregnancies among unmarried women in Malawi??

In line 364 -367, the discussion is not relevant because you indicate that after adjusting for covariates wealth status was not a significant contributor to unhealthy weight. It would be relevant if you would explain why the non-variation in unhealthy weight is observed in Malawi...You use the word Aside?? i am not sure what it means, did you mean besides?

General comments

The article with correcting minor comments is publishable.

Reviewer #2: Topic: multilevel analysis of Unhealthy body weight amin women in Malawi: Does urbanization Matter?

1) Abstract: This is well written what is important is for the authors to indicate the inter cluster correlation coefficient and indicate how it is imperative to delineate the influence across the clusters.

2) The abstract conclusions and recommendation is a bit vague, it is not capturing the key policy statement that the study is advocating o DBM. I suggest revision so that policy direct is clearly indicated and to the point.

Background

3)Line 53- requires updated reference that is recent as 2014 is way too far. I suggest revision of the reference.

Line 54-55 Requires revision as it is grammatically unbalanced. For instance the risk of dying largely occasioned by external causes of death” it is not well balance as death is already referenced and this is confusing the readers.

4)Line 59, maternal and child diseases and deaths requires revision.

5) Line 52-60. There was need for the authors to put the argument clears they are discussing unhealth body weight among women and it is not coming out clear even from a world perspective with evidence of the countries that are experiencing this situation which is reading to increased levels of morbidity and mortality among both the vulnerable youth and women. I suggest this must be incorporated.

6)Line 62, before going into Malawi, there is need for the author to justify the statement of SSA by providing a little evidence to justify such statement to highlight the gravity of the problem.

7) Line 65 and 66-The authors need to explain what the population level. The study by Bulirani et al. 2018 looked at the factors affecting nutrition status of reproductive age among women of Dedza District, Malawi was it a multi-level study or not to justify the argument? Please elaborate.

8) Line 110 and Line 120 there is need to explain the selection criteria as to how the 7231 women were identified. I propose a flow chart be drawn to outline the detail. Please adjust.

9) Line 122 and 128 This is well written and outcome variable well defined. But the emphasis shod be the dependent variable defined in the model. I consider that the one not used in the model should be deleted so that the golden thread is well pronounced.

10)Line 145 there is need to add ethnicity as to how its recoded. Make sure that all the variables used in the model are defined in this section.

11) Line 149 is the use of the chi-square appropriate for means? I propose that the use of the means is not appropriate than ordinary frequencies and percentage and the show the chi-square coefficient and p-Values.

12) Line 183 and 188 please indicate the rationale for applying the nested regression. Otherwise the results of having model 1 is not validated as the outcome variable defining the coefficient in models 2 are congruent. I would propose model 1 de dropped and concentrate at model 2 and 3. If the models 2 and 2 are included I can foresee issues of multi-collinearity with the model not converging properly. Please validate and retest the models. This is because the community model is in independent model in this case total disaggregated with the model 2. Remember the unit of analysis is always the individual levels in any multi-level analysis.

13) In the multilevel model, there is need to explain in the methodology how community poverty and education was defined. There is need for clarity. Otherwise, it is a bit confusing to a lay reader. If place of residence is dropped in the model 1, it could fit better as a community variable.

Results

14) I would like to propose that the results are presented in a mix bag If we have more than one model thus discussed, there is needs to first use within model analysis-here make sure that all significant findings are flagged. Then, the between model which now compare in relative term, the consistency variable thus causing significant variables worth highlighting. Those are the ones to be appreciated for discussion and they assist in creating a comprehensive golden thread. Try to do a self-validation on this as it will improve readability of the paper.

15) Line 393 and 394. Authors need to revisit the sentence and remove the national income. By nature DHS is a cross sectional data micro-level data set whereas national income is a macro-level data. By data science design, they are different hence the claim not justifiable. Similarly, DHS has dietary data which could be defined either directly or as a proxy and can be used. Hence, this sentence require heavy revision is at all deleted completely as it is countering some important data etiquette.

16) Line 400 and Line 408. There is need to isolate the clearly the variables that are significant when estimating the two dependent variables under discussion.

17) There is need to create an evidence based recommendation and this must be based on the consistency findings of the study. In that tertiary result review, there is need to have have a statement that is to stimulate policies. Like the last sentence in the conclusion section is out of context and require extensive review.

6. PLOS authors have the option to publish the peer review history of their article (what does this mean?). If published, this will include your full peer review and any attached files.

Reviewer #1: **Yes: **Dr Mpho Keetile

Reviewer #2: No

---

## [Author Response · Author response to Decision Letter 0]

10 Mar 2021

The comments are insightful and enrich the quality of the manuscript

---

## [Editor Report · Decision Letter 1]

16 Mar 2021

Multilevel analysis of unhealthy bodyweight among women in Malawi: does urbanisation matter?

PONE-D-20-37443R1

Dear Dr. Afolabi,

We’re pleased to inform you that your manuscript has been judged scientifically suitable for publication and will be formally accepted for publication once it meets all outstanding technical requirements.

Kind regards,

Kannan Navaneetham, PhD

Academic Editor

PLOS ONE
---

## [Editor Report · Acceptance letter]

17 Mar 2021

PONE-D-20-37443R1 

Multilevel analysis of unhealthy bodyweight among women in Malawi: does urbanisation matter? 

Dear Dr. Afolabi:

I'm pleased to inform you that your manuscript has been deemed suitable for publication in PLOS ONE. Congratulations! Your manuscript is now with our production department. 

Kind regards, 

on behalf of

Professor Kannan Navaneetham 

Academic Editor

PLOS ONE